# Structural Insights into the Chaperone-Assisted Assembly of a Simplified Tail Fiber of the Myocyanophage Pam3

**DOI:** 10.3390/v14102260

**Published:** 2022-10-14

**Authors:** Zi-Lu Wei, Feng Yang, Bo Li, Pu Hou, Wen-Wen Kong, Jie Wang, Yuxing Chen, Yong-Liang Jiang, Cong-Zhao Zhou

**Affiliations:** School of Life Sciences, Division of Life Sciences and Medicine, University of Science and Technology of China, Hefei 230026, China

**Keywords:** cryo-EM structure, cyanophage, tail fiber assembly, molecular chaperone, protein complex

## Abstract

At the first step of phage infection, the receptor-binding proteins (RBPs) such as tail fibers are responsible for recognizing specific host surface receptors. The proper folding and assembly of tail fibers usually requires a chaperone encoded by the phage genome. Despite extensive studies on phage structures, the molecular mechanism of phage tail fiber assembly remains largely unknown. Here, using a minimal myocyanophage, termed Pam3, isolated from Lake Chaohu, we demonstrate that the chaperone gp25 forms a stable complex with the tail fiber gp24 at a stoichiometry of 3:3. The 3.1-Å cryo-electron microscopy structure of this complex revealed an elongated structure with the gp25 trimer embracing the distal moieties of gp24 trimer at the center. Each gp24 subunit consists of three domains: the N-terminal α-helical domain required for docking to the baseplate, the tumor necrosis factor (TNF)-like and glycine-rich domains responsible for recognizing the host receptor. Each gp25 subunit consists of two domains: a non-conserved N-terminal β-sandwich domain that binds to the TNF-like and glycine-rich domains of the fiber, and a C-terminal α-helical domain that mediates trimerization/assembly of the fiber. Structural analysis enabled us to propose the assembly mechanism of phage tail fibers, in which the chaperone first protects the intertwined and repetitive distal moiety of each fiber subunit, further ensures the proper folding of these highly plastic structural elements, and eventually enables the formation of the trimeric fiber. These findings provide the structural basis for the design and engineering of phage fibers for biotechnological applications.

## 1. Introduction

Bacteriophages, which specifically infect the bacterial hosts, are the most abundant entities on the planet [1], and have been widely studied as antibacterial agents [2] or applied to food quality control [3]. Approximately 95% of known phages are tailed phages, which comprise an icosahedral or prolate capsid containing a double-stranded DNA genome linked to a tail that delivers the genomic DNA into the host cell [4]. The initial attachment of tailed phages to the host cells is generally mediated by specific interactions between the phage tail fibers and host surface receptors [5]. For the myophages that have a contractile tail and a baseplate of complex structure, the attachment of a myophage via tail fiber to the host cell surface often triggers the conformational transition of the baseplate, leading to sheath contraction and eventually the ejection of phage genomic DNA into the host [6,7,8].

The phage tail fibers identified to date are elongated trimeric proteins that contain distinct domains, featuring intertwined extended regions and triple β-helices [9,10,11]. As a well-investigated model of myophage, the *Escherichia coli* phage T4 possesses two kinds of fibers: the long tail fibers (LTFs) composed of four different protein components and the short tail fibers (STFs) of gp12 protein [12,13]. Given the high specificity of phage tail fibers toward the host, the tail fiber is a key apparatus for adaptation to different hosts; thus, any modifications on or mutations in the tail fibers during evolution will obviously impair the phage infection, leading to the so-called host resistance or altering the host range of phages [14]. Therefore, elucidation the mechanisms of tail fibers assembly and specificity towards the host receptors is crucial for phage therapy and other phage-based biotechnologies [15,16]. As reported previously, the distal moiety of gp37 in T4 LTF mediates receptor binding, and the proper folding of gp37 trimer requires two chaperones, gp57a and gp38, which are not incorporated into mature phage particles [17]. In contrast, the chaperones of phages Ur-λ [18] and Mu [19] have been found to exist in mature phages. Most of these chaperones belong to the large family of tail fiber assembly proteins (TFAs) that assist the correct folding and polymerization of tail fibers [20]. As the only structure-known member of TFAs, the N-terminal domain of phage Mu TFA consists of a central four-stranded β-sheet packing against a pair of α-helices on one side, in addition to an α-helix and two-stranded β-sheet on the other side [19]. It remains in the mature phage via binding to the distal moiety of the tail fiber to ensure its correct assembly and trimerization, therefore also contributing to recognizing the host receptors. Although TFAs are widespread in diverse phages, little is known on their structures and functions.

As widely distributed phages that infect the ancient cyanobacterial hosts, cyanophages can modulate the seasonal succession of cyanobacteria in aquatic ecosystems and therefore represent a potential environmentally friendly biological agent for controlling cyanobacterial blooms [21,22]. Engineering of cyanophage tail fibers could improve the efficiency of host recognition or expand the host ranges, which is the key point for synthetic engineering and environmental applications. The previous cryo-electron tomography (cryo-ET) studies showed that the tail fibers of cyanophage P-SSP7 adopt various conformations upon infection to the host [23]. The tail fiber of cyanophage A-1(L) was found to bind to the extracellular lipopolysaccharide (LPS) of the host *Anabaena* PCC 7120 [24]. However, very limited information is available for the cyanophages, which largely precludes the understanding of the molecular mechanism of cyanophage infecting the host.

In this study, we investigated the structures of the tail fiber and its chaperone of the *Myoviridae* cyanophage Pam3, which was isolated from the Lake Chaohu in China [25]. Pam3 has a minimal baseplate of only six protein components, and it possesses a much simpler tail fiber composed of only one protein gp24, providing an ideal model for synthetic engineering and biotechnological applications. Assembly of gp24 requires a specific chaperone gp25, but the underlying mechanism of how gp25 assists the assembly of gp24 remains unknown. Here, we solved the cryo-electron microscopy (cryo-EM) structure of gp24 complexed with gp25 (gp24-gp25) and revealed a modular structure of gp25 required for gp24 assembly and trimerization. Combined with in vitro biochemical assays, we propose a mechanism for tail fiber assembly and maturation processes assisted by the chaperone, which provides the structural basis for further design and engineering of cyanophages for biotechnological applications.

## 2. Materials and Methods

### 2.1. Cloning, Protein Expression and Purification

The genes *gp24* and *gp25* were amplified by polymerase chain reaction (PCR) from the genomic DNA of Pam3. The complete *gp24-gp25* DNA segment was cloned to a modified pET-28a vector with an N-terminal His_6_-tag on *gp24*. The individual *gp24* or *gp25* gene was also cloned in the pET-28a vector with an N-terminal His_6_-tag (Appendix A). Proteins were expressed in *E. coli* strain BL21 (DE3) (Novagen, Darmstadt, Germany) in LB medium containing 30 μg/mL kanamycin at 37 °C. When the optical density of cells at 600 nm (OD_600_) reached 0.8–1.0, protein expression was induced with 0.2 mM isopropyl β-D-thiogalactoside (IPTG) for 1 h at 37 °C. The cells encoding gp24-gp25 were harvested by centrifugation at 8000× *g* for 3 min and were resuspended in the binding buffer (20 mM Tris-HCl, pH 7.0, 100 mM NaCl and 7 mM β-Mercaptoethanol). Then, the cells were lysed using 30 min of sonication followed by the centrifugation at 12,000× *g* for 15 min. The supernatant containing the target proteins was transferred to a Ni-NTA column (GE Healthcare, Chicago, IL, USA) that was pre-equilibrated with binding buffer. The protein impurity was removed with 50 mM imidazole in binding buffer. The target protein was eluted with 500 mM imidazole, and further purified by size-exclusion chromatography using a HiLoad 16/600 Superdex 200 pg column. The purity of proteins was assessed using SDS-PAGE (the concentration of stacking gel is 5% and the separating gel is 4–15%) and stored at −80 °C for further use. The protein marker is GoldBand Plus 3-color Regular Range Protein Marker (YEASEN, Shanghai, China).

### 2.2. Oligomeric State Analysis

The molecular weight of gp24-gp25 complex and gp24 in solution were determined by size exclusion chromatography with multi-angle light scattering (SEC-MALS). The Superdex 200 Increase 10/300 GL column (GE Healthcare, US) pre-equilibrated with binding buffer was connected to the DAWN HELEOS II light scattering detector (Wyatt Technology, Santa Barbara, CA, USA) and the Optilab T-rEx refractive index detector (Wyatt Technology). Protein samples (1 mg/mL, 100 μL) were eluted through the column, and the results were processed and analyzed using ASTRA 7.0.1 software (Wyatt Technology). The final results were analyzed and plotted using Origin 2022.

### 2.3. Thermal Stability Assays

The purified gp24 proteins were concentrated to 2 mg/mL. Each protein sample of 15 µL was heated for 5 min at various temperatures from 57 to 87 °C with an interval of 2 °C by PCR Amplifier (Bio-Rad, Hercules, CA, USA) and centrifuged at 12,000× *g* for 10 min at 4 °C to remove the precipitated proteins. The protein concentration in the supernatant was determined using a NanoDrop (Thermo Fisher Scientific, Waltham, MA, USA). The proteins retained in the supernatant were compared with the unheated sample to calculate the soluble proportion at each temperature. The data points were fitted with a Boltzmann equation to calculate the Tm value using Origin 2022.

### 2.4. Cryo-EM Sample Preparation, Data Collection and Processing

The purified gp24-gp25 proteins were concentrated to 4.7 mg/mL and incubated with 0.03% (*v*/*v*) NP40 (Nonidet P40). The 3.5 µL protein samples were loaded on a 300-mesh carbon-coated golden grid (Quantifoil R1.2/1.3, Großlöbichau, Germany) and flash frozen in liquid ethane using the Vitrobot (FEI). Grids were blotted for 8 s and incubated for 30 s with a blot force of −2 in 100% relative humidity at 8 °C. After removing the excess sample using the filter paper, the golden grid was quickly transferred to liquid nitrogen for storage.

The cryo-EM datasets were collected using an FEI Titan Krios 300 kV transmission electron microscope equipped with the K3 electron-counting K3 detector at the Cryo-EM Center at University of Science and Technology of China. A total of 1,435 movies (30 frames, each 0.12 s, total dose 55 e/Å^2^) were collected with a magnification of 120,000× in the super resolution mode with a defocus range of −1.5 to −2.5 μm. More than 959,630 particles were automatically picked by using cryoSPARC [26]. After initial 3D classification by imposing C1 symmetry, the remaining ~80% particles (~320,000) representing the gp24-gp25 complex were subject to the 3D classifications by C3 symmetry. Finally, a total of 243,989 good particles were selected for the 3D refinement via imposing the C3 symmetry, yielding a 3.1 Å cryo-EM map for the gp24-gp25 complex at 3:3 stoichiometry (Appendix A).

The structures of gp24 and gp25 were predicted using AlphaFold2 (version 2.1) [27] by importing the FASTA sequences to the Genetic Database. The output models with the highest score were selected as initial models to individually fit the cryo-EM map using the UCSF Chimera [28]. The initial models were manually adjusted using COOT [29], and were then automatically refined using Real_space_refine in PHENIX [30]. After several rounds of iterations of the manual and automatic refinements, the final model was validated by PHENIX and MolProbity [31]. All structural figures were drawn by PyMOL (https://pymol.org, accessed on 15 May 2022). The statistics for cryo-EM data collection and processing, model building and refinement were listed in Table 1. The interface analysis of the gp24-gp25 complex was performed by PDBsum [32] and manually checked by PyMOL. The distance within 3 Å between the main-chain amide and carbonyl groups of neighboring GRMs were predicted as the hydrogen bonds.

### 2.5. Bioinformatics Analyses

The molecular functions of gp24 and gp25 were predicted by BLASTp (https://blast.ncbi.nlm.nih.gov, accessed on 12 May 2021) and HHpred [33] using the primary sequences as the search hint. All the parameters for the BLAST and HHpred search of gp24 or gp25 were set as the default values. The database of non-redundant protein sequences is used for BLAST search, and PDB_mmCIF70, Pfam-A_v35, UniProt-SwissProt-viral70, NCBI Conserved_Domains v3.19 are applied for HHpred analysis. The proteins with functional annotation in the top hits were selected for further analysis. The primary sequence alignments were performed using Multalin [34]. The structural similarity search was performed using the DALI server [35].

## 3. Results

### 3.1. The Chaperone gp25 Forms a Stable Complex with the Tail Fiber gp24 at a Stoichiometry of 3:3

Bioinformatics analysis indicated that gp24 shows 29% sequence similarity (10% sequence identity) to the long tail fiber tip protein gp38 of phage S16. By contrast, the BLASTp search of gp25 gave hits of previously identified tail chaperones, such as tail assembly proteins (TACs) and TFAs. Moreover, HHpred prediction showed that gp25 shares a high similarity to the TFA of phage Mu. Mass spectrometry analysis of the intact Pam3 virions detected the gp24 protein, but not gp25 (Appendix A), further suggesting that only gp24 is present in the mature Pam3 particles. Altogether, it indicated that gp24 and gp25 of Pam3 encode the tail fiber protein and its chaperone, respectively.

To investigate the molecular functions of gp24 and gp25, we individually expressed His_6_-tagged full-length gp24 (His-gp24), gp25 (His-gp25) or co-expressed His-gp24 with gp25 in *E. coli*. Overexpression of individual gp24 yields a large fraction of recombinant proteins in inclusion bodies, leaving a small fraction of soluble proteins, which showed two peaks in size-exclusion chromatography corresponding to the trimeric and monomeric gp24, respectively (Figure 1a,b). The chromatographic profile of His_6_-gp25 also shows two peaks which correspond to the trimeric and monomeric gp25, respectively (Figure 1a,b). Co-expression of gp24 and gp25 yields a large amount of soluble proteins, which are homogeneous as shown by a uniform peak and a small shoulder in size-exclusion chromatography (Figure 1a). It implied that gp25 is required for the proper folding and trimerization of gp24, similar to that in phage T4 [17,20] and Mu [19]. The following sodium dodecyl sulfate polyacrylamide gel electrophoresis (SDS-PAGE) analysis of this uniform peak reveals a complex composed of gp24 and gp25 whereas the shoulder peak corresponds to the gp24 trimer (Figure 1b). Further size-exclusion chromatography with multi-angle light scattering (SEC-MALS) analysis of gp24 and gp24-gp25 complex obtained the apparent molecular masses of 94 and 129.4 kDa, respectively (Figure 1c,d), which are comparable to the theoretical molecular weight of the trimeric gp24 (84 kDa) and gp24-gp25 (135 kDa). It demonstrated that gp24 and gp25 form a stable complex at a stoichiometry of 3:3. Notably, the purified gp24 proteins present as trimers in the SDS-PAGE when the samples were not boiled (Appendix A), suggesting that it is rather stable and shows resistance to the detergent SDS, consistent with its high melting temperature (Tm) of 76.6 °C (Appendix A). The high Tm values have also been observed in other tail fiber proteins, such as T4 long tail fiber tip protein gp37, T4 short tail fiber protein gp12 [36] and *Salmonella* phage P22 tail spike [37].

### 3.2. Three gp24 Subunits Form a Simplified Trimeric Tail Fiber

To gain structural insights into gp25-assisted assembly of gp24 trimer, we determined the cryo-EM structure of the gp24-gp25 complex at 3.1 Å resolution (Figure 2a, Appendix A, Table 1). The residues from Leu31 to Pro280 of gp24 were modeled to the final structure (Figure 2b). By contrast, the structure of the full-length gp25 (residues Met1~Phe162) was solved (Figure 2c). In the gp24-gp25 complex, the gp24 trimer adopts an extended structure, whereas three separated β-sandwich domains of gp25 trimer embrace the distal moieties of gp24 trimer at the center, forming an elongated complex structure of ~140 Å in length (Figure 2a). Seeing along the three-fold axis, the gp25 trimer covers the distal loops of gp24, making these loops inaccessible to the solvent. These loops located at the distal of tail fiber were proposed to recognize the host; thus, the chaperone gp25 should be released prior to the maturation of viral particles.

Three subunits of gp24 form an elongated trimeric structure of ~97 Å in length, resembling a lighted torch (Figure 2a). Each gp24 subunit consists of three distinct domains: the α-helical domain forming the shaft of the torch, the TNF-like and glycine-rich domains constituting the flame of the torch (Figure 2b). Each pair of subunits in the gp24 trimer is stabilized by extensive interactions, forming a total buried interface area of ~1500 Å^2^, which accounts for more than a quarter of the total surface area of each subunit. Compared to the loose contact within the pairwise TNF-like domain and glycine-rich domain, extensive hydrophobic interactions in the N-terminal α-helical bundle contribute the majority to the trimerization of gp24.

The N-terminal α-helical domain of gp24 comprises of two α-helices, a long α1 and a short α2. In the trimeric gp24, the three long helices assemble into an intertwined α-helical bundle that docks tail fiber to the baseplate. The three short helices pack against the C-terminal ends of the long helices and form a joint to link the TNF-like domains. The TNF-like domain, which is structurally similar to the tumor necrosis factor-α (TNF-α) monomer [38], with a root-mean-square deviation (RMSD) of 2.3 Å over 53 Cα atoms. They both possess two parallelly aligned five-strand β-sheets; however, compared to the TNF-like domain of gp24, the human TNF-α has an extra short helix. These β-strands are stretched along the fiber axis, and fold together to form a glycine-rich domain at the distal end of the fiber (Figure 3a). In total, 39% residues in this domain are glycines (49/125 residues), which consists of six extended loops (termed L1~L6). Each loop has two glycine-rich motifs (GRMs) that adopt an elongated conformation (Figure 3a). These loops run parallel to the fiber axis and stack together into a four-layer lattice at an interval distance of ~5 Å (calculated from Cα atoms) between adjacent GRMs (Figure 3a). Extensive main-chain hydrogen bonding networks among the GRMs assist in maintaining the compact structure of the glycine-rich domain.

Further analysis of these GRMs revealed a similar conformation and organization for each GRM (Figure 3a). Ramachandran plot analysis revealed that the dihedral angles of each GRM are close to β-sheet. Among the six loops, the loops L1 and L6 that sits at the center of the glycine-rich domain, have the longest and most glycine-rich stretches with the conserved motif of -GGGGxGG- (Figure 3b). Notably, the sequence diversity gradually increases from the proximal to the distal end of the GRMs, forming a much less conserved distal tip of glycine-rich domain. A couple of polar residues at the distal ends of GRMs were proposed to form the receptor-binding sites that determine the host specificity. Overall, the glycine-rich domain presents an unusual architecture consisting of a conserved glycine-rich flexible stem and a hypervariable distal tip that recognizes diverse host receptors (Figure 3a).

Structural similarity search using DALI [35] revealed that the structure of gp24 resembles the tail fiber adhesin tip protein gp38 of Salmonella phage S16 (PDB 6F45) [39], with a RMSD of 2.7 Å over 52 Cα atoms. They both have three distinct domains but differ greatly in structure (Figure 4a). There are differences between the middle TNF-like domain of gp24 and the β-helix domain of S16 gp38. Despite sharing a similar architecture, the glycine-rich domains differ in the number of GRMs, where Pam3 gp24 has 12 GRMs forming a four-layer lattice, compared to a three-layer lattice formed by 10 GRMs of S16 gp38. In addition, the N-terminal α-helical bundle of gp24 is much longer than that of S16 gp38, which contains only three short α-helices that attach to gp37 through a hydrophobic platform. Moreover, the TNF-like and glycine-rich domains of gp24 are structurally similar to the ligand-binding extracellular domain of Caenorhabditis elegans anaplastic lymphoma kinase (PDB 7LIR) [40], with an RMSD of 2.6 Å over 125 Cα atoms. The TNF-like domain of gp24 also closely resembles the β-sandwich domain that mediates oligomerization of the eukaryotic TNF [37]. Altogether, the similarities on the overall architecture, especially the glycine-rich domain required for host recognition, strongly indicated that phage tail fibers might have evolved from a common ancestor [41,42,43].

### 3.3. Three gp25 Subunits Assemble into a Trimeric Chaperone

Each gp25 subunit consists of two distinct domains: the N-terminal β-sandwich domain (residues Met1~Ala55) and the C-terminal α-helical domain (residues Pro64~Phe162) connected by an eight-residue linker. The β-sandwich domain has two β-sheets composed of two and three antiparallel β-strands, respectively, in addition to a short helix α1, whereas the α-helical domain consists of four α-helices (α2~α5) and an η-helix (Figure 2c). In gp25 trimer, the three α-helical domains assemble into a helical bundle that mediates the trimerization of gp25. The three helices α3 form an inner core of the helical bundle, via extensive hydrophobic interactions in addition to nine pairs of hydrogen bonds (Appendix A). By contrast, the three β-sandwich domains are separated from each other, which individually protrude outwards like three “tentacles” to stick onto the junction between C-terminal TNF-like and glycine-rich domains of gp24 trimer.

Structural similarity search demonstrated that the β-sandwich domain of gp25 is most structurally similar to the TFA N-terminus of phage Mu (Tfa_Mu_) (PDB 5YVQ), with an RMSD of 3.0 Å over 48 Cα atoms. Superposition of gp25 β-sandwich domain against Tfa_Mu_ showed that the central β-sheet is well aligned, whereas the surrounding α-helices vary greatly (Figure 4b). Compared to Tfa_Mu_, the structure of gp25 is simpler and lacks three α-helices (corresponding to α1, α3 and α4 in Tfa_Mu_), which contribute to tail fiber binding in phage Mu [19]. As the C-terminal domain of Tfa_Mu_ was not solved, we predicted its structure by AlphaFold2 [27]. It has two α-helices, one of which could be superimposed onto the helices α2 and α3 of gp25, whereas the other helix corresponds to α4 of gp25 (Figure 4b). Consistent with the previous finding [19], we propose that the C-terminal domain of Tfa_Mu_ might also form a helical bundle that mediates the trimerization to ensure the correct assembly of trimeric tail fiber. Similar to Tfa_Mu_, the chaperone gp25 also adopt a modular structure in which the non-conserved N-terminal β-sandwich domain binds to the tail fiber, whereas the conserved C-terminal α-helical domain mediates the trimerization and assembly of tail fiber.

In addition, the overall structure of gp25 is also similar to the intramolecular chaperones located at their C-termini of phage tail fibers (Appendix A), including the L-shaped tail fiber of *E. coli* phage T5 (PDB 4UW8) [44], the tail spike protein of *E. coli* phage K1F (PDB 3GW6) [45] and the neck fiber gp12 of *Bacilllus* virus GA1 (PDB 3GUD) [45]. They all adopt a similar architecture that consists of a compact α-helical bundle and three individually extended “tentacles” (Appendix A). However, despite similar in the α-helical bundle, gp25 differs greatly from these intramolecular chaperones in the “tentacle”. Compared to the segments of extended loops or antiparallel β-strands in intramolecular chaperones, gp25 possesses an individual β-sandwich domain. Nevertheless, the overall architecture similarity suggests that these chaperones function in a similar way to assist the correct folding and oligomerization of client proteins.

### 3.4. The Chaperone gp25 Drives the Correct Folding and Assembly of gp24 Tail Fiber

In the complex structure of gp24-gp25, the trimeric gp25 embraces the gp24 trimer mainly via the N-terminal β-sandwich domains (Figure 2a). The β-sandwich domain of one gp25 subunit interacts with the surface-exposed regions at the lateral of the TNF-like and glycine-rich domains of one gp24 subunit, forming an interface area of ~550 Å^2^. The interactions are mainly mediated by β2 and α1 of gp25, and β5~β7 of gp24, forming extensive hydrophobic interactions, in addition to seven pairs of hydrogen bonds (Figure 5a). The electrostatic surface potential showed that the interfaces between gp24 and gp25 are somewhat complementary in shape and surface potential (Figure 5b).

The present complex structure provides insights into the assembly process of gp24 trimer, assisted by the chaperone gp25. We propose that the chaperone gp25 assemble into a trimer via the C-terminal α-helical domains, from which three N-terminal β-sandwich domains protrude outwards and function as three “tentacles” to search the nascent gp24 subunits. Once binding to the highly flexible C-terminal region of gp24, the chaperone protects these intertwined elements of gp24 and eventually enable their proper folding. Notably, the N-terminal α-helical domain of the tail fiber could trimerize as well in the absence of the chaperone. Once the trimeric tail fiber is correctly folded and registered, the chaperone should be released, most likely thanks to a relatively small interface area between gp24 and gp25 in addition to an eight-residue flexible linker that bridges the two domains of the chaperone.

## 4. Discussion

In this study, we report the cryo-EM structure of the simplified tail fiber complexed with its chaperone from the myocyanophage Pam3, which provides insights into the assembly mechanism of the tail fiber assisted by an intermolecular fiber chaperone. The tail fiber gp24 adopts an intertwined and elongated trimeric structure with a modular organization different from the previously reported ones. The gp24 has a TNF-like domain of two parallelly aligned β-sheets that is rarely found in phages. The TNF-like domain stretches out and forms a four-layer structure of the glycine-rich domain harboring 12 GRMs, which is the most prominent feature of gp24. Compared to a much more complex assembly of tail fibers in other myophages, such as T4 [13] and S16 [39], the Pam3 tail fiber contains only one protein gp24. Moreover, the structure of gp24 is also simplified, which contains only three distinct domains. By contrast, other phage fibers usually contain extra domains, such as the needle-like domain of the intertwined triple β-helix or triple β-spiral folds. Therefore, the tail fiber of Pam3 is relatively simple, which might become an ideal model for future applications in synthetic biology.

A key finding of our study is that the full-length structure of the chaperone gp25 adopts a modular organization. The chaperone gp25 is composed of two separate domains: the N-terminal β-sandwich domain responsible for binding to the tail fiber, and the C-terminal α-helical domain mediates the trimerization. This modular organization is commonly found in other protein chaperones, such as phage Mu TFA [19] and RuBisCO accumulation factor Raf1 [46]. Sequence analysis showed that the N-terminal domain of gp25 is variable, which coincides with the diverse structures of phage tail fibers. By contrast, the C-terminal domain of gp25 is relatively conserved, which shows a clear similarity in sequence to the C-terminal region of Tfa_Mu_ and other Tfa proteins. These results suggest the intermolecular chaperone of tail fiber also adopts a modular organization, a totally distinct N-terminal domains in combination with a highly conserved C-terminal domain.

In conclusion, our complex structure of gp24-gp25 provides insights into the assembly mechanism of phage tail fiber assisted by an intermolecular chaperone. Structural analysis clearly shows that the role of chaperone is to initiate the correct assembly of tail fiber at the distal moiety, thus constraining the intertwined structural elements to be orderly aligned, which provides a model for the action of tail fiber chaperones. Our studies might also pave the way for further understanding and synthetic engineering of myophages with a simplified tail fiber.

## Figures and Tables

**Figure 1 viruses-14-02260-f001:**
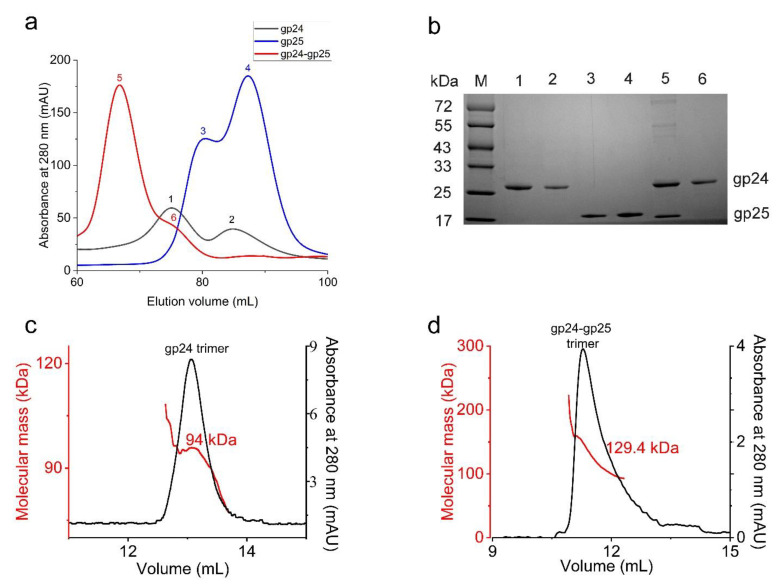
The purification of gp24, gp25 and the gp24-gp25 complex. (**a**) The size-exclusion chromatography profiles of gp24 (black line), gp25 (blue line) and gp24-gp25 complex (red line) using a Superdex 200 Increase 10/300 column. The corresponding peaks are labeled sequentially (1~6). (**b**) The SDS-PAGE analysis of the peaks from **a**. The samples were boiled for 10 min before loading to the SDS-PAGE. The gp25 protein in the gp24-gp25 complex has no tag, whereas the gp25 alone contains a His_6_-tag. (**c**,**d**) show the SEC-MALS analysis of the gp24 trimer and the gp24-gp25 complex. The black line represents the absorbance at 280 nm (*Y*-axis on the right), and the red line indicates the molecular weight (*Y*-axis on the left).

**Figure 2 viruses-14-02260-f002:**
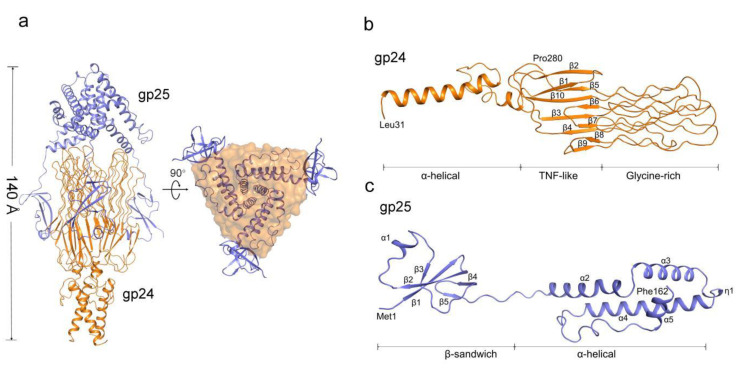
Overall structure of the gp24-gp25 complex. (**a**) Overall structure of the gp24-gp25 complex shown in side and bottom views. The structure of gp25 is shown in cartoon and colored blue. The structure of gp24 is colored orange and shown in cartoon and surface, respectively, in the two views. Cartoon representation of one subunit of gp24 (**b**) and gp25 (**c**). The distinct domains, the secondary structural elements and the terminal residues are labeled.

**Figure 3 viruses-14-02260-f003:**
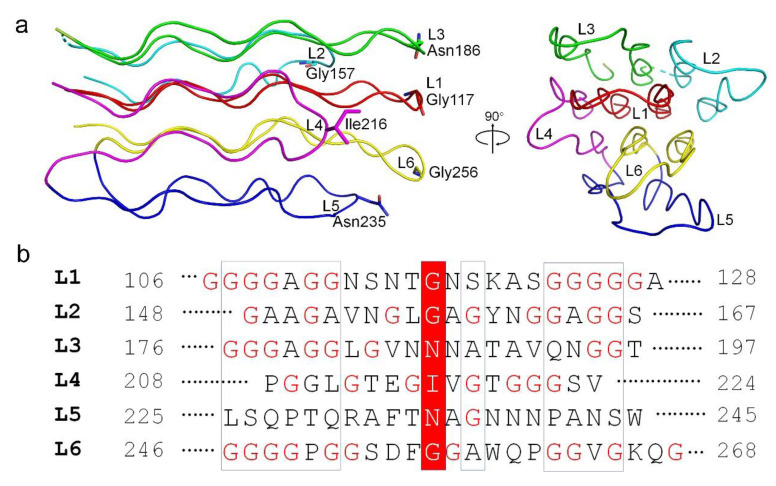
The organization of the glycine-rich domain of gp24. (**a**) The structural arrangement of the six loops, shown in side and top views. Each loop is colored differently. The residues at the outermost of each loop are shown in sticks and labeled. (**b**) Sequence alignment of the six loops (L1~L6) in the glycine-rich domain of gp24. The residues at the distal end of each loop are highlighted in a red box and the glycine residues are marked in red. The numbering of each loop is labeled.

**Figure 4 viruses-14-02260-f004:**
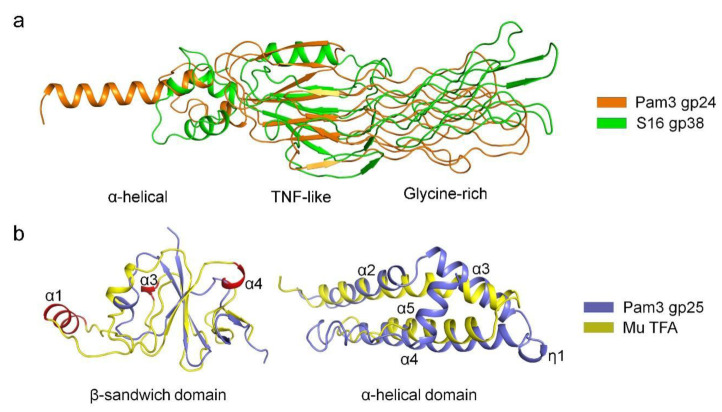
Structural comparisons of gp24 and gp25 with homologs. (**a**) Superposition of Pam3 gp24 against gp38 of phage S16 (PDB 6F45). The structures are shown in cartoon, and colored orange for gp24 and green for gp38. The three domains of gp24 are labeled. (**b**) Superposition of the two domains of Pam3 gp25 against those from phage Mu TFA (left panel for the β-sandwich domain and right panel for the α-helical domain). The model of the C-terminal domain of phage Mu TFA was predicted by AlphaFold2 (right panel). The structures are shown in cartoon and colored differently, with the secondary structural elements of gp25 labeled sequentially. The additional helices α1, α3 and α4 of the N-terminal domain of phage Mu TFA are colored red and labeled.

**Figure 5 viruses-14-02260-f005:**
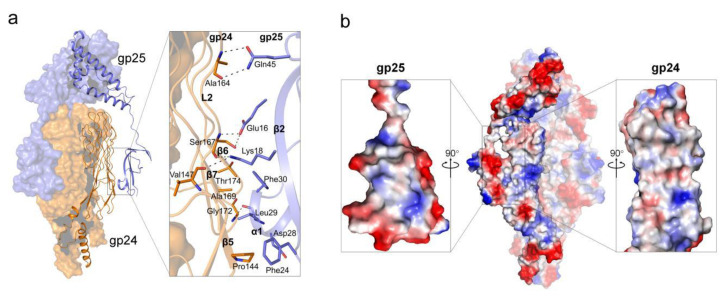
The interface between gp24 and gp25. (**a**) The interaction pattern between gp24 and gp25. The interface residues are shown as sticks and the hydrogen bonds are represented by black dashed lines. (**b**) The electrostatic surface potential of gp24-gp25 trimer. red, negative charged; white, neutral charged; and blue, positive charged regions (±50 kT/e). The cut-open views of the interfaces from gp25 (left) and gp24 (right) are shown as the inlets.

**Table 1 viruses-14-02260-t001:** Cryo-EM parameters, data collection and refinement statistics.

Data Collection and Processing	gp24-gp25 (PDB 7YPX) (EMD-34017)
Magnification	120,000
Voltage (keV)	300
Electron exposure (e^-^/Å^2^)	55
Pixel size (Å)	1.07
Defocus range (µm)	−1.5~−2.5
FSC threshold	0.143
Map resolution range (Å)	2.14~999
Symmetry imposed	C3
Initial particle images (no.)	959,630
Final particle images (no.)	243,989
Map resolution (Å)	3.1
Refinement	
Real-space correlation coefficient	0.84
Initial model used	Ab-initio
Map sharpening B factor (Å^2^)	−161.1
Model composition	
Non-hydrogen atoms	8808
Protein residues	1236
RMS deviation from ideality	
Bond lengths (Å)	0.004
Bond angles (°)	0.578
Validation	
MolProbity score	1.86
Clash score	12.71
Poor rotamers (%)	0.23
Ramachandran plot	
Favored regions (%)	96.3
Allowed regions (%)	3.7
Outliers (%)	0 0

## Data Availability

Accession Numbers. The coordinates for the gp24-gp25 complex have been deposited in the Protein Data Bank (PDB) under the accession code 7YPX. The cryo-EM map of gp24-gp25 has been deposited to the EMDB database under the accession code EMD-34017.

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
