# Peer review of "Structural Insights into the Chaperone-Assisted Assembly of a Simplified Tail Fiber of the Myocyanophage Pam3"

_viruses, 2022, doi:10.3390/v14102260_

Round 1

Reviewer 1 Report

Zhou et al. describe the structural characterization of phage Pam3 tail fiber (gp24) complex formation mediated by phage chaperone gp25 using size exclusion chromatography, TEM, and AphaFold. Gp25 forms a complex with gp24 with a 3:3 stoichiometry, where gp25 is at the periphery and gp24 is at the center. Through comparative analysis to homologous proteins of other phages, the authors suggest mechanism of action and evolutionary origins for this class of proteins (tail fiber assembly proteins: TFA), which little is known. A key finding of this study, is that gp25 is composed of two domains, N-terminal domain, which is responsible for binding to the tail fiber, and the C-terminal domain, which mediates trimerization and assembly of gp24. Structural insights into phage tail fiber is broadly important, as this information can be used for engineering phage host range for developing antibacterial agents for treating multidrug resistant infections or food quality control. Overall, the paper is well written with clean figures. I suggest publication with minor revisions.

Minor comments:

·      Page4, line 149. Specify % homology

·      Add Mass Spec data showing gp25 is not in mature virion.

·      Figure 1B. Instead of showing MW on y-axis, can you show on secondary x-axis?

·      Page 5, line 170. Add what theoretical MW to discussion.

·      Page 6, line 208. How similar?

·      Page 6, line 227, please cite Figure 3a.

Reviewer 2 Report

In the manuscript entitled "Structural Insights into the Chaperone-Assisted Assembly of a Simplified Tail Fiber of the Myocyanophage Pam3" by Zi-Lu Wei et al., the authors gain insight into the structural determinants responsible for the proper architecture of the complex formed by the Pam3 phage fibre tail and its chaperone counterpart. Biochemical analysis showed that the gp25 chaperone is important for the solubility of its substrate, the gp24 Pam3 fibre tail, supporting its role as a chaperone. Authors solve the structure of gp24-gp25 complex at 3.1 Å in which a gp25 trimer protects loops from gp24 trimer lacking secondary structure elements that otherwise most probably promote the protein aggregation and transfer to inclusion bodies. The solved structure led the authors to propose an assembly mechanism that, in my view, lacks some data to be supported and I believe the sequential events described in the authors' assembly model are not supported by the presented data. Therefore, the assembly model and the data concerning the oligomerization state of initial assembly stages need to be revised. From my point of view, data is also compatible with a model in which gp24 and gp25 form trimers previously to the formation of the complex.

I describe below the drawbacks that, in my opinion, are present in this work and that need to be resolved in order to accept the article for publication:

- Figure 1, red trace, shows a large peak corresponding to the gp24-gp25 trimer complex (labelled as 1). However, a shoulder starts to appear behind at the position of the gp24 trimer (red trace Figure 1A), indicating a small fraction of the total protein that does not contribute to the complex. What is the oligomeric state of this solubilised protein? Does it appear a peak corresponding to gp25 trimer. dimer o monomer? What is the chromatographic profile of gp25 alone? Data supporting the presence of gp25 monomers, dimers or trimers would help to clarify the question regarding oligomerisation states. Please provide a complete chromatographic profile of the gp24-gp25 complex and gp25 alone.

- Indicate molecular mass of the proteins. The molecular mass of the band adscribed to gp24 trimer (~65 kDa) seems lower than expected (~80 kDa), being approximately twice that of the monomer. It may be a dimer and not a trimer. Also, the gel conditions should be indicated in Materials and Methods. Has gp25 been expressed and analysed (SDS-PAGE and SEC-MALS) individually?

- It is difficult to understand the reason why gp24 does not apper in its monomeric form after heating for the SDS-PAGE while the Tm was found to be ~77 ºC. Under denaturing conditions during preparation of sample for gel electrophoresis, it should be dissociated and run in its monomeric form. If it is able to precipitate, it should form quite large aggregates that couldn’t properly run in the gel. Please, clarify this issue.

- The imposed C3 symmetry restricts the number of particles one fourth, from ~1000000 to ~250000. Could it occur that trimeric form of the complex, which has been structurally solved, only represents a small fraction of all possible oligomeric states? What would happen if no symmetry group is imposed? Is there another population of complexes with different stoichiometry? Although less probable, this possibility could explain the observed discrepancy on molecular mass and the fact that gp24 had to be modelled.

- As I understand, modelled gp24 was used as template to be fitted in trimeric densities. However, in the presence of gp25 and due to its chaperone activity, one would expect to find a stable complex with proper fold. Can the authors discuss on the reason why gp24 was not solved?

Minor:

- Please, clarify what is the architecture of the clones used to express gp24 and gp25. Is there a unique His-tag? Can gp24 and gp25 be expressed separately or are they in same vector His6-gp24-gp25? A scheme of the vectors’ architecture in the Supplementary file would help to the reader.

- Please, include a brief description on the AlphFold procedure used to predict gp24 structure.

- Include references, or methods, in regard to the source of sequence alignments concluding that gp24 and gp25 have the functions described.

- Protein denaturation curves would be clearer if total amount of protein is expressed in abscissa axis. Can the authors plot a graph showing [Soluble protein] vs T? How much protein remain in the soluble fraction?

- Lines 215-216. “Extensive main-chain hydrogen bonding networks among the GRMs assist in maintaining the compact structure of the glycine-rich domain”. How did the authors carry out the contact analyses? Please, include it in Material and Methods.

- Lines 322-325. I do not understand the reasoning that the small interface and flexibility of the linker region of gp25 promotes the release of gp24. This cannot be extracted from statically averaged cryo-EM densities. Can the authors expand on this reasoning?

- Figure 4, line 262. In the figure legend, it says that “The domains of gp24 al labelled”, but there is no labels on the figure. Please, include these labels in the figure.

- Line 285. “contribute to tail fibre binding in phage Mu.” Please, add a corresponding reference.

- Are glycine-rich loops conserved alongside other phages? What is the goal of sequence analysis between loops within the same phage? Please, explain further this issue.

- I did not find any reference to VideoS1 in the text. Is there any video? I was not able to find any video. Please, clarify.

Typos:

- Line49: assemble to assembly

- Line 186. “The residues from Leu31 to Pro280 of gp24 was modeled to the final structure”. Should say “The residues from Leu31 to Pro280 of gp24 were modeled to the final structure”.

Round 2

Reviewer 2 Report

The new version of the manuscript has been improved and the conclusions nuanced, which was my main disagreement. In conclusion, I am in favour of accepting the work for publication in Viruses in the present form.